# A general synthesis of azetidines by copper-catalysed photoinduced anti-Baldwin radical cyclization of ynamides

Clément Jacob[1,2], Hajar Baguia[1], Amaury Dubart[1], Samuel Oger[1], Pierre Thilmany[1], Jérôme Beaudelot[1,3], Christopher Deldaele[1], Stefano Peruško[1,2], Yohann Landrain[1], Bastien Michelet [1], Samuel Neale [4], Eugénie Romero[1], Cécile Moucheron [3✉], Veronique Van Speybroeck [4✉], Cédric Theunissen [1✉] & Gwilherm Evano [1✉]

A general anti-Baldwin radical 4-exo-dig cyclization from nitrogen-substituted alkynes is reported. Upon reaction with a heteroleptic copper complex in the presence of an amine and under visible light irradiation, a range of ynamides were shown to smoothly cyclize to the corresponding azetidines, useful building blocks in natural product synthesis and medicinal chemistry, with full control of the regioselectivity of the cyclization resulting from a unique and underrated radical 4-exo-dig pathway.

[1] Laboratoire de Chimie Organique, Service de Chimie et PhysicoChimie Organiques, Université libre de Bruxelles (ULB), Avenue F. D. Roosevelt 50, CP160/ 06, 1050 Brussels, Belgium. [2] Organic Synthesis Division, Department of Chemistry, University of Antwerp, Groenenborgerlaan 171, 2020 Antwerp, Belgium. [3] Laboratoire de Chimie Organique et Photochimie, Service de Chimie et PhysicoChimie Organiques, Université libre de Bruxelles (ULB), Avenue F. D. Roosevelt 50, CP160/08, 1050 Brussels, Belgium. [4] Center for Molecular Modeling, Ghent University, Tech Lane Ghent Science Park Campus A, Technologiepark 46, 9052 Zwijnaarde, Belgium. ✉email: Cecile.Moucheron@ulb.be; veronique.vanspeybroeck@ugent.be; Cedric.Theunissen@ulb.be; Gwilherm.Evano@ulb.be

**U**p to 90% of naturally occurring organic molecules contain either a carbocycle or a heterocycle[1], a figure that goes up to 97% in the drug industry[2]. Methods enabling straightforward and efficient syntheses of cyclic organic molecules are therefore of utmost importance and it is not surprising that intramolecular cyclizations have been extensively studied over the years, resulting in the development of remarkably efficient reactions for the synthesis of a broad range of carbo- and heterocycles. For these transformations to be of any synthetic usefulness, their regioselectivity must be fully controlled. This can be predicted, with more or less precision, by the iconic Baldwin's rules for ring closures[3], rules that have been comprehensively revisited[4,5] since their publication in 1976. While most cyclization modes are well documented, some of them still remain rather elusive, notably for the formation of small ring systems such as four-membered ones. Among these, the 4-exo-dig cyclization is certainly the least-explored one, despite a strong potential. It was indeed originally predicted, based on an acute angle of attack of the three interacting atoms, to be unfavourable by Baldwin, while the complementary 5-endo-dig was proposed to be favourable (Fig. 1a)[3]. This proposed preference for the endo attack in digonal cyclizations is in sharp contrast with the general preference for radical exo-cyclizations suggested by Beckwith[6] and which was later on revised, notably with the remarkable contributions of the Alabugin group who demonstrated that digonal exo-cyclizations should be favoured[7]. However, examples of such cyclizations are still limited. Although few cases of anionic[8–12] and formal palladium-[13–15] or gold-[16] catalysed 4-exo-dig cyclizations have been described, only two examples of radical ones have been reported by the Malacria[17] and Kambe[18] groups, from quite specific substrates, while the 5-endo-dig ring closure has been shown to be favoured in some cases (Fig. 1b)[19]. A general 4-exo-dig radical cyclization therefore remains untapped and meeting this challenge would be of high significance in terms of synthetic chemistry and, from more fundamental points of view, especially in terms of reactivity.

Based on the peculiar reactivity of ynamides[20,21] and their ability to participate in radical processes[22,23], we hypothesized that these nitrogen-substituted alkynes might provide an excellent opportunity for the development of such a process. The presence of the nitrogen atom in between the radical centre and the triple bond could facilitate the 4-exo-dig cyclization by bringing the reacting centres in proximity, polarizing the triple bond and stabilizing the resulting vinyl radical (Fig. 1c). In addition to the potential development of a general anti-Baldwin 4-exo-dig cyclization, this would also provide a versatile entry to highly substituted azetidines, an important scaffold found in a range of natural products and bioactive molecules (representative examples shown in Fig. 1d)[24,25]. Due to the unique 3D shape of these small nitrogen heterocycles and the range of biological activities they display, azetidines have been extremely studied as active pharmaceutical ingredients recently and their synthesis has been therefore intensively reinvestigated in recent years[26–30].

We report in this manuscript the development of a general anti-Baldwin radical 4-exo-dig cyclization of ynamides, which are readily activated using copper-based photoredox catalysis. A variety of highly functionalized azetidines are obtained in good yields and they are shown to be useful and versatile building blocks which can further participate in various post-functionalization reactions. DFT calculations also indicate that the 4-exo-dig cyclization process is kinetically favoured over the more common 5-endo-dig pathway.

## Results

**Feasibility and optimization.** To design an attractive and efficient synthesis of azetidines and test the feasibility of a radical 4-exo-dig cyclization of ynamides, we opted for a photoredox approach to generate the radical species required for the cyclization. In particular, we turned our attention to copper-mediated photoredox catalysis[31–33] and [Cu(bcp)DPEphos]PF$_6$[34] was first

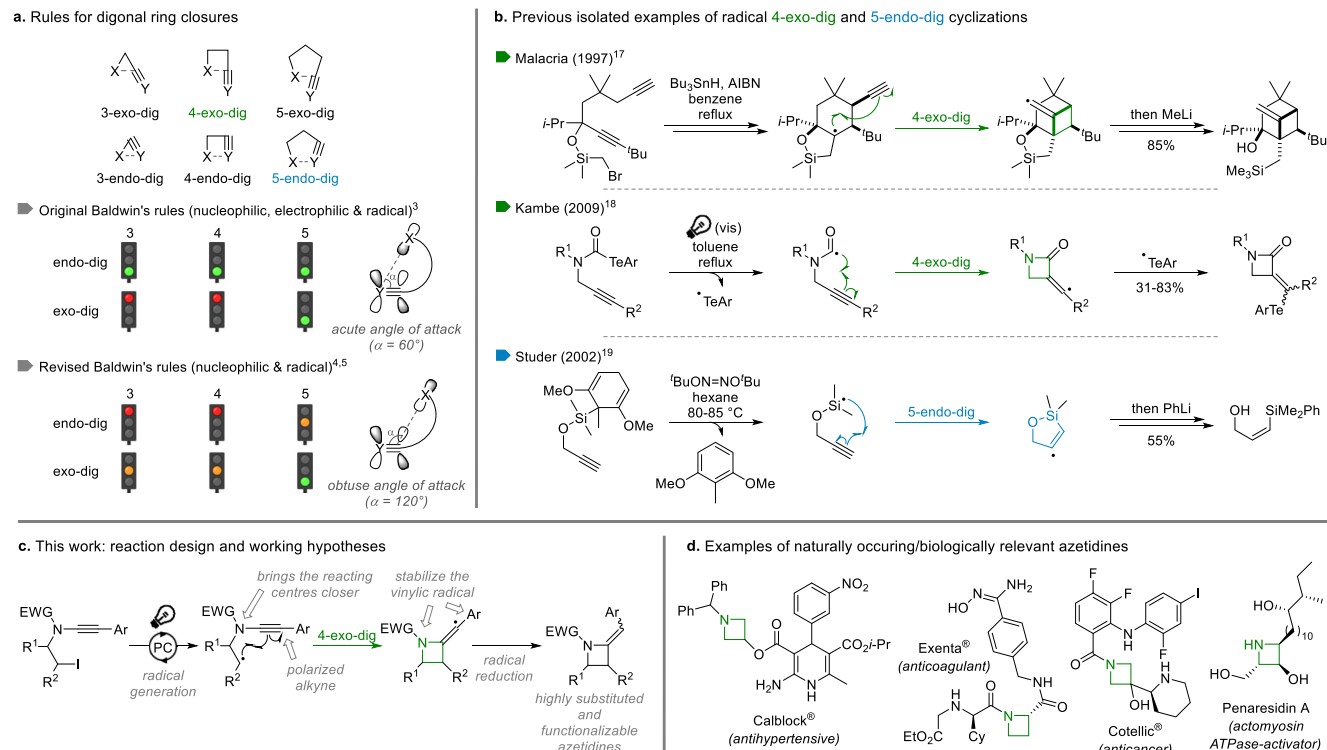

**Fig. 1 Digonal ring closures for the synthesis of azetidines. a** Original and revised Baldwin's rules for digonal ring closures. **b** Rare examples of radical 4-exo-dig and 5-endo-dig cyclizations. **c** Reaction design: radical 4-exo-dig cyclization of ynamides triggered by visible-light activation with a photoredox catalyst (PC). **d** Representative examples of naturally occurring/biologically relevant azetidines.

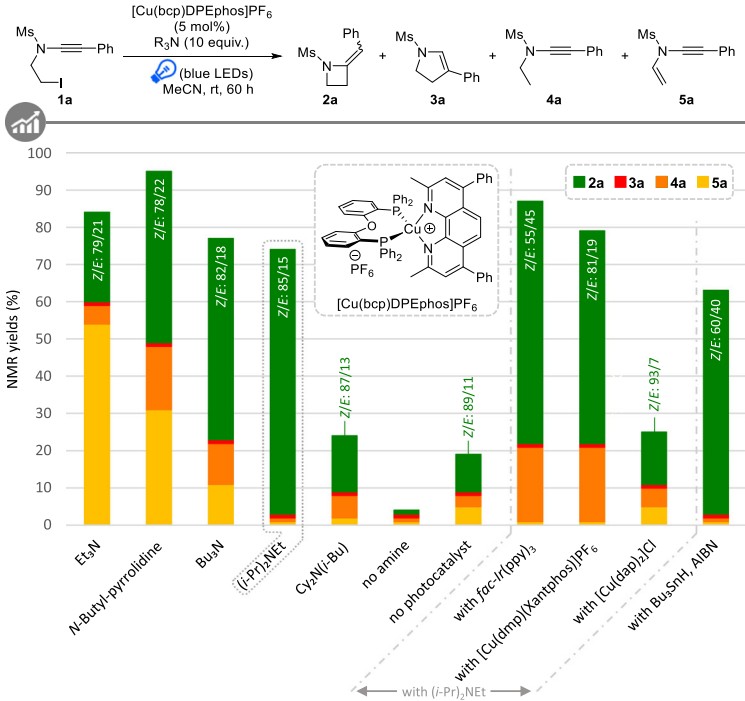

**Fig. 2 Optimization of the 4-exo-dig radical cyclization of ynamides to azetidines.** Reactions performed on a 0.2 mmol scale. NMR yields and *Z/E* ratios were determined by ¹H NMR analyses of the crude reaction mixtures using mesitylene as internal standard.

evaluated as it already proved efficient for the activation of organic halides[35]. With this goal in mind, *N*-iodoethyl-ynamide **1a** was reacted with [Cu(bcp)DPEphos]PF₆ in the presence of a range of tertiary amines as sacrificial reductants, under irradiation with blue LEDs in acetonitrile at room temperature. As evidenced with selected results summarized in Fig. 2, our working hypothesis turned out to be relevant since the desired azetidine **2a** was formed in most trials and the regioselectivity of the radical cyclization was found to be totally in favour of the 4-exo-dig cyclization, as no traces of dihydropyrrole **3a** resulting from the complementary 5-endo-dig process were observed in crude reaction mixtures. The nature of the tertiary amine was found to have a dramatic impact on the efficiency of the cyclization: while triethylamine displayed only moderate efficiency and mostly promoted the elimination of **2a** to **5a**, a noticeable improvement was observed when switching to *N*-butyl-pyrrolidine, tributylamine and Hünig's base, the latter being the most efficient and resulting in the formation of 71% of **2a** with only traces of reduced product **4a**. The use of a less soluble amine such as dicyclohexylisobutylamine[35] proved detrimental, although a sacrificial amine was strictly required as a redox-neutral atom-transfer radical cyclization was found to be inoperative. While a slow and inefficient background light-induced cyclization of ynamide **1a** was observed in the absence of a photoredox catalyst, the use of [Cu(bcp)DPEphos]PF₆ is however required to accelerate the reaction and achieve synthetically useful yields. Importantly, no reaction was observed in the absence of light. Noteworthy, *fac*-Ir(ppy)₃, one of the most efficient photoredox catalysts to date, displayed a slightly lower efficiency for this cyclization while other copper-based photoredox catalysts also proved to be less efficient, the reduction to **4a** becoming again a serious side-reaction in these cases, and the use of classical tin-based conditions was less efficient. In all trials, azetidine **2a** was formed as a mixture of *Z* and *E* isomers in variable ratios (from 55/45 to 93/7) but always in favour of the *Z* isomer.

Importantly, we verified at this stage that the nature of the system utilized for the irradiation of the reaction mixture only

had a little impact on the outcome of the cyclization: the use of blue LED strips, a photoreactor (LZC-CCP-4V) with irradiation at 420 nm or a parallel photoreactor (PhotoRedOx Box) with a blue Kessil lamp gave close yields and comparable *Z/E* ratios as exemplified by the cyclization to **2a** with all systems. Noteworthy, the activation of other carbon–halogen bonds could also be demonstrated with the successful cyclization of an *N*-bromoethyl-ynamide **1a_Br** providing 55% of the desired azetidine **2a** with a *Z/E* ratio of 88/12 (Fig. 3) while an *N*-chloroethyl-ynamide **1a_Cl** was however found unreactive. Finally, the nature of the protecting group on the nitrogen atom was also briefly probed and the cyclization was found to be still operative when replacing the mesyl group (Ms) by a Boc group, although with a slightly reduced efficiency, azetidine **2b** being isolated in 59% yield (Fig. 3).

**Scope and limitations**. With these optimized conditions in hand, we next investigated the scope of this reaction by first evaluating the influence of the nature of the ynamide R³ substituent. As demonstrated by results summarized in Fig. 3, a range of aryl groups are tolerated, and the cyclization is not strongly affected by their electronic properties since electron-rich (**2c–f,i–k**) and electron-deficient (**2l–n**) aromatic substituents provide the corresponding azetidines in fair to good yields and in comparable *Z/E* ratios. Their steric properties were shown to have a limited impact on the efficiency of the process, similar yields being obtained with *para*- (**2c**), *meta*- (**2d**) and *ortho*- (**2e**) tolyl groups. Notably, a bromide and a chloride are well tolerated, as demonstrated with the cyclization to **2l** and **2m**, which provides opportunities for further structural diversification. Importantly, heteroaryl-substituted ynamides are also amenable to the radical cyclization, furyl- and thiophenyl-substituted azetidines **2o** and **2p** being obtained in 61% and 57% yields, respectively. Switching to non-aromatic substituents revealed the requirement for a radical-stabilizing group on the starting ynamide, TIPS-substituted azetidine **2q** being obtained in reduced yield (25%)

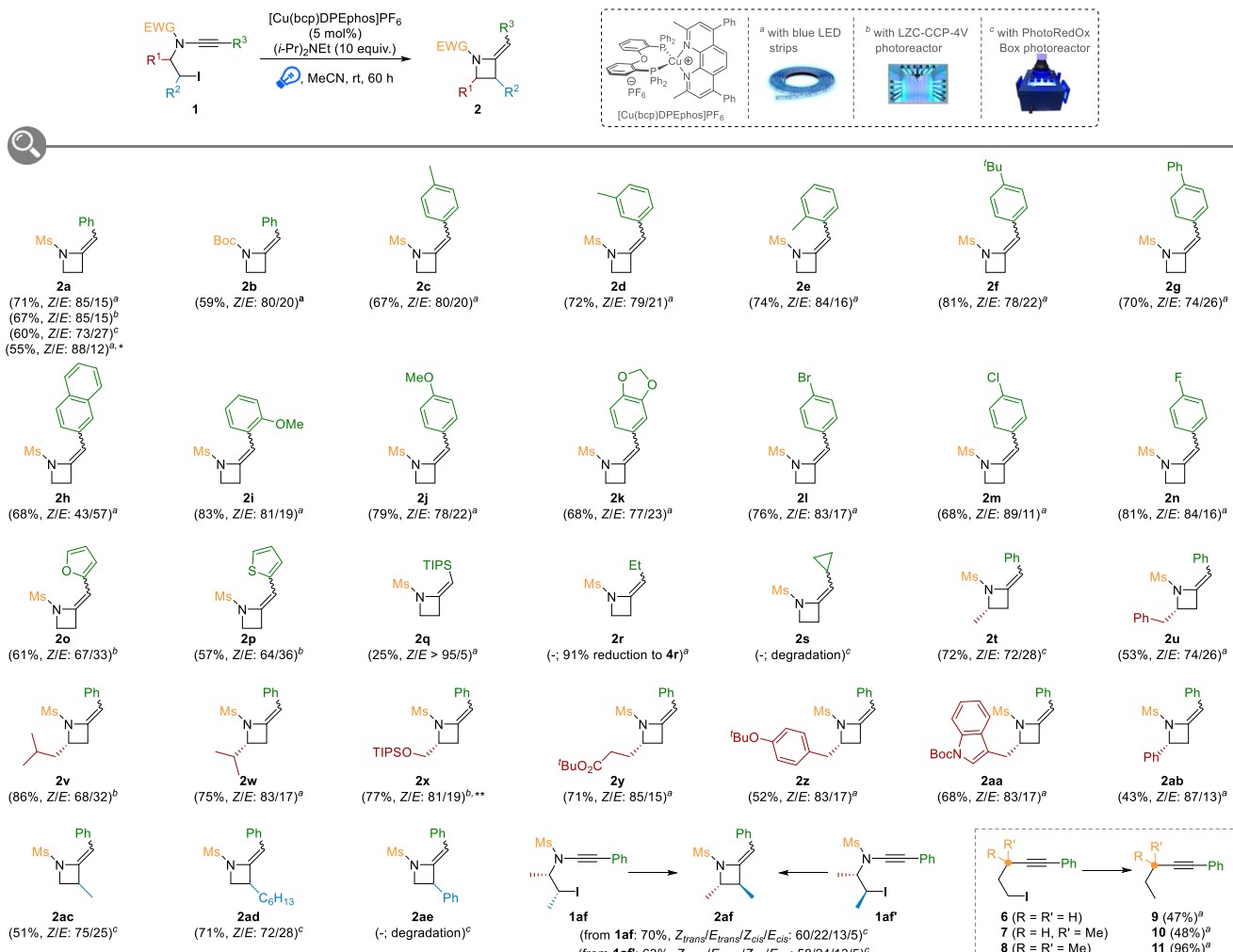

**Fig. 3 Scope of the 4-exo-dig radical cyclization of ynamides to azetidines.** Reaction conditions: ynamide **1** (0.2 mmol), [Cu(bcp)DPEphos]PF$_6$ (5 mol%), (*i*-Pr)$_2$NEt (2 mmol), MeCN (0.1 M), rt, 60 h. Isolated yields are provided and *Z/E* ratios were determined by [1]H NMR analyses of the crude reaction mixtures. * from *N*-bromoethyl-ynamide **1a$_{Br}$**. ** during 120 h.

while the presence of ethyl or cyclopropyl groups totally inhibited the cyclization, in the latter case, products resulting from a radical cyclization/ring opening being undetectable in the crude reaction mixture.

After investigating the influence of the R$^3$ substituent on the outcome of the radical 4-exo-dig cyclization of ynamides to azetidines, we next moved to the study of the scope of the reaction with respect to the substitution of the iodoethyl chain by varying the R$^1$ and R$^2$ groups. The starting ynamides **1** being prepared by N-alkynylation of suitably protected aminoalcohols[36,37], their radical cyclization is therefore especially suitable for the synthesis of chiral non-racemic azetidines **2t-ab** that can be obtained in good yields from a range of commercially available aminoacid-derived aminoalcohols. This also further highlights the efficiency and functional group tolerance of our radical cyclization since protected alcohols, esters and hetero-arenes do not interfere with the outcome of the process. C3-Substituted azetidines can also be efficiently obtained by placing the substituent β to the nitrogen atom in the starting ynamides **1**, as shown with the synthesis of methyl- and hexyl- substituted azetidines **2ac** and **2ad**. One limitation of the cyclization was found in an attempt to prepare C3-phenyl-azetidine **2ae**, the benzylic radical generated by photoinduced activation of the corresponding ynamide being certainly too stabilized for the

cyclization to occur. A variety of optically enriched β-aminoalcohols possessing two stereogenic centres α and β to the nitrogen atom being commercially and/or readily available, the radical cyclization of ynamides derived from these substrates enables the preparation of C2,C3-disubstituted azetidines such as **2af**. In this case, starting from both diastereoisomers **1af** and **1af'** logically gave the corresponding azetidine **2af** in similar yields and selectivities, the *trans* relationship between the two methyl groups in **2af** being preferred.

Finally, and as an important note, the nitrogen atom in the starting γ-iodo-alkynes was shown to be crucial for the cyclization to occur since replacing the protected amine in **1** by various methylene groups in **6–8** resulted in exclusive reductions, despite the strong Thorpe–Ingold effect that could be expected with substrate **8**.

In order to push forward and further test the efficiency of this azetidine synthesis in "real-life" situation, the radical cyclization was performed from much more complex substrates (Fig. 4). In this perspective, phenylalanine- and estrone- containing yna-mides **1ag** and **1ah** were subjected to our standard reaction conditions: even with these challenging substrates, the radical cyclization proceeded efficiently, structurally complex azetidines **2ag** and **2ah** being obtained in 63% and 59% yields, respectively, showcasing the robustness of the cyclization. Gratifyingly, the

**Fig. 4 Radical 4-exo-dig cyclization of ynamides to azetidines with complex substrates.** Reaction conditions: ynamide **1** (0.2 mmol), [Cu(bcp)DPEphos]PF$_6$ (5 mol%), (i-Pr)$_2$NEt (2 mmol), MeCN (0.1 M), r.t., 60 h. Isolated yields are provided and Z/E ratios were determined by $^1$H NMR analyses of the crude reaction mixtures.

**Fig. 5 Cascade radical 4-exo-dig/7-endo-trig cyclization to bicyclic azetidine 2aj.** Reaction conditions: ynamide **1aj** (0.2 mmol), [Cu(bcp) DPEphos]PF$_6$ (5 mol%), (i-Pr)$_2$NEt (2 mmol), MeCN (0.1 M), r.t., 16 h. Isolated yield is provided.

cyclization is also amenable to the synthesis of spirocyclic azetidines such as **2ai** whose spirocyclic pyrrolidine-azetidine core can be found in diverse natural products and active pharmaceutical ingredients, such as the chartellines **12**[38], the chartellamides **13**[39] and Delgocitinib **14**[40], a janus kinase inhibitor developed by Japan Tobacco and recently commercialized for the treatment of autoimmune disorders and hypersensitivity, including inflammatory skin conditions.

**Cascade radical cyclization**. Having extensively explored the scope and limitations of this radical 4-exo-dig cyclization and demonstrated its full regioselectivity, we next envisioned using this strategy in cascade radical cyclizations. Iodinated enynamide **1aj**, a substrate which can undergo a total of four different radical domino 4-exo-dig/5-endo-dig and 6-exo-trig/7-endo-trig cyclizations, was therefore subjected to our standard conditions. While the reaction was found to be moderately efficient and required a shorter reaction time to reach 39% yield, which did not improve after 16 h, a single cyclized product **2aj**, resulting from an unprecedented radical 4-exo-dig/7-endo-trig cascade cyclization—the latter having also little precedents[41–46] due to competing 6-exo-trig cyclization[41]—could be isolated (Fig. 5). Byproducts resulting from other cyclization modes or competing reactivity could not be detected and the low mass balance was attributed to competing radical oligomerization of the starting ynamide.

**Post-functionalization and chemical diversification**. With an efficient entry to azetidines in hand, we next turned our efforts towards their chemical diversification, mostly aiming at demonstrating the usefulness of the exocyclic double bond as a versatile handle for the preparation of a range of polysubstituted

azetidines. With this goal in mind, a series of azetidines **2** was therefore first subjected to hydrogenation with Pearlman's catalyst which proceeded smoothly to provide the corresponding C2-substituted azetidines and, starting from chiral azetidines, with good to excellent levels of diastereoselectivity in favour of the cis isomers, as demonstrated with the hydrogenation to **15f–i** (Fig. 6a).

An ionic reduction of **2a** could also be performed using a combination of trifluoroacetic acid and deuterated triethylsilane, which enabled a clean reduction of **2a** to selectively deuterated azetidine **15a$_D$** (Fig. 6b), such deuterated compounds being of importance with the growing role of selectively deuterated small organic molecules in drug discovery[47]. In addition to the reduction of the exocyclic double bond in model azetidine **2a**, the azetinium ion involved in its ionic reduction to **15a$_D$** could also be trapped with a Grignard reagent, as exemplified with the synthesis of azetidine **16** bearing a pseudo-quaternary stereocentre, such azetidines being especially challenging to prepare with standard methods[24,25]. In a similar perspective, activation of the enamide in **2a** with bromine followed by hydrolysis provided 79% of brominated β-aminoketone **17** resulting from an electrophilic bromination of the exocyclic enamide followed by nucleophilic addition of a bromide to the transient spirocyclic bromonium ion/β-bromo-azetinium ion and hydrolysis. Moreover, we could also show that the alkene in **2a** could be cleaved by ozonolysis to afford the corresponding β-lactam **18**, the combination of this oxidative cleavage with our radical 4-exo-dig cyclization providing an efficient access to β-lactams that could be hardly obtained otherwise. The exocyclic enamide in **2a** could in addition be further arylated with diphenyliodonium triflate in the presence of catalytic amounts of copper(II) triflate[48] to azetidine **19** possessing a fully substituted exocyclic enamide moiety.

Finally, and in a last effort to address one of the main limitations of our radical 4-exo-dig cyclization producing a mixture of usually hardly separable Z and E isomers in which the Z is significantly the major one, we briefly addressed its isomerization. To our delight, simply reacting a mixture of the Z and E isomers of **2a** (Z/E: 85/15) with catalytic amounts of iodine at room temperature for 2.5 h enabled a smooth and full equilibration to the E isomer (**E**)-**2a**. This isomerization was moreover shown to be general since azetidines **2d**, **2e**, **2g**, **2l**, **2aa**, **2ab**, **2ag** and **2ah** could all be fully isomerized to their E isomers

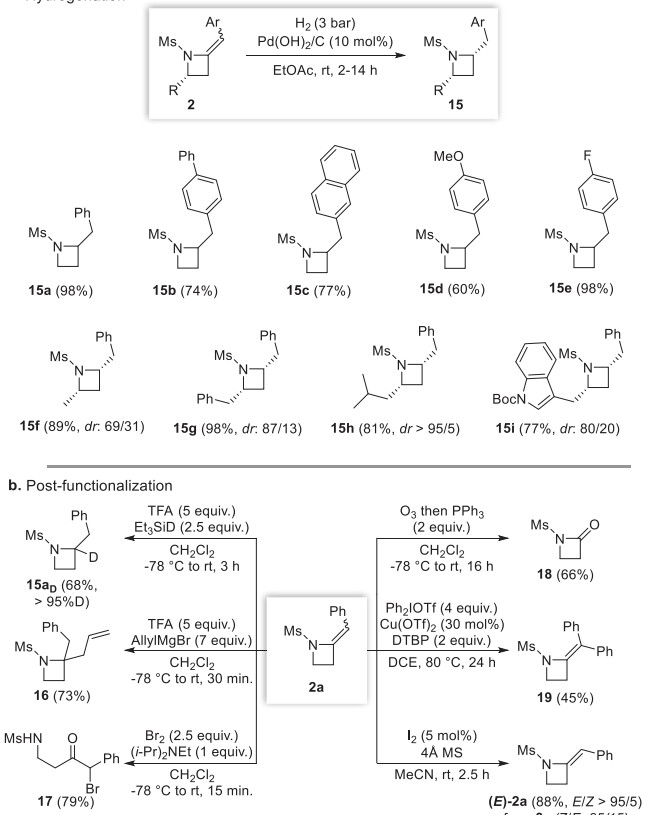

**Fig. 6 Chemical diversification of representative unsaturated azetidines.**
**a** Hydrogenation of representative azetidines. **b** Post-functionalization of the exocyclic double bond of model azetidine **2a**. Isolated yields are provided and *dr* were determined by $^1$H NMR analyses of the crude reaction mixtures. TFA trifluoroacetic acid, DTBP 2,6-di-*tert*-butylpyridine and MS molecular sieves.

in yields ranging from 80% to 88% (see Supplementary Information for details).

**Computational study and mechanistic considerations.** After demonstrating the synthetic usefulness of this cyclization of ynamides into azetidines, a computational study was undertaken on substrates **1a** and **8** in which DFT calculations were performed to understand the preference for the observed 4-exo-dig cyclization over the alternative 5-endo-dig pathway, as well as to assess the dramatic influence of the nitrogen atom and aromatic substituent of the starting ynamides on the outcome of the cyclization (see Supplementary Information for computational details). Computed free energy profiles for 4-exo-dig and 5-endo-dig cyclizations from the corresponding deiodinated alkyl radical intermediates, generated through copper-catalysed photoinduced single electron reduction of the starting iodides[35], are presented in Fig. 7.

As for ynamide **1a**, the 4-exo-dig cyclization via **TS-4$_{1a}$** (+13.4 kcal/mol) is kinetically favoured over the alternative 5-endo-dig process via **TS-5$_{1a}$** (+17.1 kcal/mol), even if the corresponding vinyl radical **5P$_{1a}$** is thermodynamically favoured over vinyl radical **4P$_{1a}$** resulting from the observed 4-exo-dig process. This is actually not entirely unexpected with regards to the lower inherent ring-strain in **5P$_{1a}$**, indicating that the cyclization leading to the desired azetidines is kinetically driven. We next focused our efforts to getting additional insights into the stereoselectivity of the cyclization, the Z isomers being

predominantly formed in all cases and analysis of the *Z/E* ratio revealing that the *Z* isomer is exclusively formed at the beginning of the reaction and then slowly equilibrates to the *E* isomer over time. Hence, the stereodivergent reductions of vinyl radical **4P$_{1a}$**, via intermolecular hydrogen atom transfer, was also characterized to probe the *Z/E* distribution and to rationalize the observed predominant formation of the *Z* isomer. The difference in intrinsic barriers for the reduction indicates a $\Delta\Delta G^{\ddagger} = -2.0$ kcal/mol in favour of the *Z* isomer, which is in good agreement with the observed selectivity and again supports a process under kinetic control, likely resulting from the reduction occurring at the less sterically hindered face of the transient vinyl radical in **TS-Z$_{1a}$** in which the bulky Ms group is facing away from the amine reductant (Fig. 7). Conversely, the *E* isomer **E$_{1a}$** (−24.8 kcal/mol) is the more thermodynamically favoured isomer of the two by 5.7 kcal/mol, which is consistent with the slow equilibration to this isomer observed over time.

Characterization of 4-exo-dig vs. 5-endo-dig cyclizations for compound **8** revealed a qualitatively consistent kinetic and thermodynamic scenario with that of ynamide **1a**, where the 4-exo-dig cyclization is kinetically favoured over the 5-endo-dig cyclization, the latter remaining however the thermodynamically favoured process. Importantly, the barrier for the 4-exo-dig cyclization of ynamide **1a** is lower than that of compound **8**, which nicely reflects the difference of reactivity observed experimentally between aromatic-substituted ynamides **1a-p,t-ad,af** and compounds **6–8**. Moreover, a comparison of the difference between 4-exo-dig and 5-endo-dig barriers in **1a** (where $\Delta\Delta G^{\ddagger} = 3.7$ kcal/mol) with that of **8** ($\Delta\Delta G^{\ddagger} = 3.9$ kcal/mol) reveals that the *N*-mesyl moiety equally accelerates both stereodivergent cyclization processes, as opposed to more favourably enhancing the 4-exo-dig process.

To further probe the underlying cause of this difference in reactivity, electronic structure analyses of **1a** and **8** were undertaken to evaluate the influence of the triple bond polarization on the outcome of the cyclization. While computed partial atomic charges somewhat reveal the polarized nature of the triple bond in ynamides **1a-ai** (see Supplementary Table 1), we also calculated the bond polarities according to the index introduced by Raub and Jansen (also referred to as the Raub–Jansen index)[49,50]. According to their definition, this index, which will hereafter be labelled as the bond-polarity index **$p_{xy}$**, calculates atomic contributions to electron densities of chemical bonds via complementary analyses of AIM[51] and ELF[52,53] electron density partitions. The triple bond polarity of **1a** (**$p_{CC} = 0.17$**) is stronger than that of compound **8** (**$p_{CC} = 0.01$**), as **1a** has the largest polarity index. This nicely reflects the reactivity of the substrates towards radical cyclization and confirms that the nitrogen atom has the most significant polarizing effect, while the phenyl group only marginally contributes to bond polarization. The role of the aromatic group is most likely to stabilize the resulting transient vinyl radical prior to onwards reduction. Indeed, evaluation of residual spin-density in the resulting transient vinyl radical species **4P$_{1a}$** and **4P$_8$** (see Supplementary Information) clearly shows this role of the aromatic group, as residual unpaired spin density is drawn from the carbon atom α to the nitrogen atom and shared across alternating C atoms in the aromatic ring. Despite this role of the phenyl group in stabilizing **4P$_8$**, the lack of reactivity of **8** itself towards 4-exo-dig cyclization demonstrates however the synergistic requirement of both the nitrogen atom and the aromatic group to promote the radical 4-exo-dig cyclization.

In conclusion, we have reported an efficient and general anti-Baldwin radical 4-exo-dig cyclization which converts readily

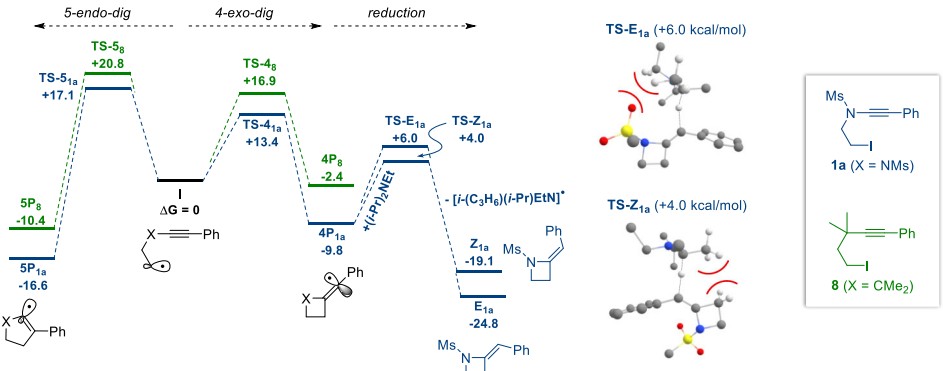

**Fig. 7 Computational study.** Computed free energy profiles of 4-exo-dig, 5-endo-dig and reduction (**1a** only) of **1a** (blue) and **8** (green) from the respective radical species **I** at the M052X-D3,MeCN/def2-QZVPP/M052X-D3/def2-TZVP level of theory. Also shown are the structures of **TS-E$_{1a}$** and **TS-Z$_{1a}$** where differences in clashing of the vinyl radical and the amine reductant are highlighted to rationalize the kinetic predominance of Z isomer formation (hydrogens not involved in reduction or primarily in the clashing between the vinyl radical and amine are omitted for clarity).

available ynamides into the corresponding azetidines using a heteroleptic copper-based photoredox catalyst. Notable features of this cyclization are its total regioselectivity, its broad substrate scope, as well as the opportunities it offers for further functionalization of the targeted azetidines. Besides providing a versatile access to highly functionalized azetidines, a structural motif of ever-growing importance in medicinal chemistry and natural product synthesis, and useful insights into the radical chemistry of ynamides, this reaction also highlights the efficiency of copper-based photoredox catalyst [Cu(bcp)DPEphos]PF$_6$, which represents an interesting alternative to iridium and ruthenium complexes. Further studies to broaden the scope of photoredox transformations available with copper-based photoredox catalysts are currently running and will be reported in due time.

## Methods

**General procedure for the cyclization of ynamides to azetidines**. An oven-dried vial was charged with [Cu(bcp)DPEphos]PF$_6$ (5 mol%) and the iodoethyl-ynamide **1** (1 equiv.). The vial was fitted with a rubber septum, evacuated under high vacuum and backfilled with argon. Freshly distilled and degassed acetonitrile (0.1 M) and distilled *N,N*-diisopropylethylamine (10 equiv.) were then added and the reaction mixture was stirred under visible-light irradiation (using a LCZ-CCP-4V photoreactor and 420 nm tubes) or under blue LED irradiation (using either blue LED strips or an EvoluChem™ PhotoRedOx Box and a blue Kessil LED lamp) for 60 h at r.t. The reaction mixture was then filtered through a pad of Celite® (washed with Et$_2$O) and concentrated under reduced pressure. The crude residue was finally purified by flash column chromatography over silica gel to afford the desired azetidine **2**.

## Data availability

The authors declare that all the data supporting the findings of this study are available within the paper and its Supplementary Information files or from the corresponding authors upon request.

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

## Acknowledgements

Our work was supported by the Université libre de Bruxelles (ULB, ARC grant ENLIGHTEN ME, C.M. and G.E.), the Federal Excellence of Science (EoS) programme (BIOFACT, Grant No. O019618F, V.V.S. and G.E.) and the Région de Bruxelles Capitale—Innoviris (2019-BRIDGE-5 PhotoCop, C.M. and G.E.). H.B., A.D., P.T., C.D., Y.L., and J.B. acknowledge the Fonds pour la formation à la Recherche dans l'Industrie et dans l'Agriculture (FRIA) for graduate fellowships. C.T. acknowledges the FNRS for a "Chargé de Recherche" fellowship. S.N. and V.V.S. acknowledge funding from the Research Board of the Ghent University (BOF) and the Research Foundation —Flanders (FWO). The computational resources and services used were provided by Ghent University (Stevin Supercomputer Infrastructure) and the VSC (Flemish Super-computer Center), funded by the Research Foundation—Flanders (FWO). We are grateful to Dr. Gaël De Leener and Mrs. Axelle Bourez (ULB) for their efficiency and assistance with HRMS analyses.

## Author contributions

C.M., C.T., and G.E conceived and designed the experiments. S.N. and V.V.S. conceived and performed the calculations. C.J., H.B., A.D., S.O., P.T., J.B., C.D., S.P., Y.L., B.M., E.R., and C.T. performed the experiments and participated in discussions. G.E. and C.T. prepared the manuscript. S.N. and V.V.S. wrote the computational sections of the manuscript.

## Competing interests

The authors declare no competing interests.

## Additional information

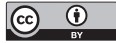

