## [Peer Review File · Nature Communications]

A General Synthesis of Azetidines by Copper-Catalyzed Photoinduced anti-Baldwin Radical Cyclization of YnamidesReviewers' Comments:

Reviewer #1:

Remarks to the Author:

The manuscript titled "A General Synthesis of Azetidines by Copper-Catalyzed Photoinduced anti-Baldwin Radical Cyclization of Ynamides" by Jacob, Baguia, Dubart, Oger, Thilmany, Beaudelot, Deldaele, Perusko, Michelet, Neale, Romero, Moucheron, Van Speybroeck, Theunissen, and Evano describes the development of a radical cyclization approach to azetidines via a regioselective 4-exo-dig cyclization of nitrogen-substituted alkynes. The introduction to the manuscript clearly defines the new fundamental discovery in terms of cyclization technology, as well as the need for more selective ways to prepare and install azetidines. The scope and limitations of the new method are nicely described and the experimental procedures for starting material preparation and the highlighted cyclization are comprehensively covered in the Supporting Information. The computational studies provide kinetic reasoning to support the observed experimental trends and some of the derivatization studies, such as isomerization to the E-isomer, are well-selected.

This manuscript describes a useful new synthetic process with interesting mechanistic implications that has been presented in a scholarly manner and merits publication in a top tier journal such as J. Am. Chem. Soc. or Angew. Chem. Int. Ed. However, the manuscript falls somewhat short of the benchmarks of the wide-readership of Nat. Comm. due to the lack of examples of the new method being used to access specific scaffolds found in target molecules or tested on more complex starting materials with potential competing reactivity. Addition of these examples, could make the manuscript suitable for publication in Nat. Comm.; otherwise, a more focused top tier chemistry journal is more appropriate.

Some additional comments and considerations for the authors and editors.

1. The cascade reaction shown in Figure 4 is a highlight of the manuscript but could be more informative if the authors also disclosed the corresponding observed competing reactivity. It is important that the authors are able to isolate 23% of the single product but it would be interesting to know what other pathways are accessible and present opportunities for further optimization.
2. Is the iodine-promoted E/Z isomerization general for a range of azetidine products? The moderate dr is one of the drawbacks of Figure 3 but if the iodine isomerization is a general solution, this could address the alkene dr concern regarding Figure 3.
3. The authors may want to reconsider the use of "peculiar" in the introduction. While it is unlikely that the authors intended the somewhat negative connotation of "peculiar", substitution with "specialized" or "specific" may be more appropriate in describing the work of Malacria and Kambe.
4. The spectra included in the Supporting Information is extensive, but many of the ¹H NMR spectra of the azetidines have obvious impurities in the baseline. For a large subset of these compounds, it seems like the concentration of the NMR sample is particularly dilute and that impurities in the NMR solvent rather than the compound itself are being observed. The authors should retake the ¹H NMR spectra for compounds such as 2a, 2d, 2ab, where the baseline is indicating a dilute sample, and perform additional purification of compounds such as 2e, 2k, 2aa, where the impurities seem more likely to be coming from the sample rather than the solvent.

Reviewer #2:

This manuscript by Jacob and coworker reports a new methodology of making azetidines allowing for a copper/photoredox catalysis via the challenging 4-exo-dig radical cyclization. The reaction proceeds with *N*-iodoethyl-ynamide and furnishes the corresponding azetidine products including further functionalization in fair to good yield. A wide range of scope is provided, and the methodology is extended to radical cascade cyclization via an interesting 4-exo-dig/7-endo-trig, and a single cyclized product is afforded. Furthermore, a deep mechanistic rationalization involving DFT computational work was conducted to support the regioselectivity of the radical cyclization. This is a significant progress in copper/photoredox catalysis radical cyclization from both synthetic and computational standpoints. Therefore, I recommend its publication in Nat. Comm. after revisions. Specific comments and questions are provided below:

1. The reactions seem to work only with *N*-iodoethyl-ynamide, and failed when replacing the protected amine by various methylene groups. What happens if replacing by dimethylmalonate?
2. The reactions work great with aromatic-substituted (R^3) ynamide. What happen with other radical stabilized groups, such as ester- and boronate-substituted, etc.
3. In Figure 3, the authors said the radical cyclization was inhibited with the presence of cyclopropyl group (**2s**). Did the authors observe any cyclization/ring-opening product? Although the authors showed product **2af** from compound **1af** and **1af'** afforded the same diastereoselective, the ring-opening is also a strong evidence to support the radical pathway.
4. The authors should explain how to determine the stereochemistry of compound (**Z**)-**2a**, (**E**)-**2A**, and **2b**, etc. by NOESY in SI.
5. The authors should provide ^{19}F NMR in SI.
6. The NMR of compounds **1af'** and **16** look impure.
7. Based on the computed energetics for 4-exo-dig cyclization, **1r** has a similar barrier as **1a**, and is also thermodynamically favorable, albeit with smaller thermodynamic drive (-5.3 vs -9.8 kcal/mol). Meanwhile, the 5-endo-trig cyclization of **1r** is even smaller in barrier (16.5 kcal/mol) compared to **1a** (17.1 kcal/mol), and the resulting products are in similar energy levels. These results seem unable to explain the experimental results of **1r** being inactive or undergoing reduction instead of cyclizations. What is the energetics for **4P_{1r}** undergoing following reduction? This might provide some insight into the reactivity of **1r**.

Reviewer #3:

Remarks to the Author:

The authors present a well-conducted study on the photochemical cyclization of ynamides that lead to azetidines. The results obtained are certainly an interesting contribution to the growing toolbox of light-induced, radical cyclizations, and moreover, a useful class of compounds is accessible by this approach. The supporting information makes a good impression, giving credit to the work.

Nevertheless, I am not convinced that the study presented meets the requirements for publication in Nat. Commun. By now it is recognized that 4-exo-dig cyclizations cannot really be considered to be an exception to the Baldwin rules (work by Gilmore). Moreover, the work by Kambe (reference 18) reports a quite related cyclization leading to very similar products as reported in this study. The design principle by the authors, i.e. biasing the conformation by an N-substituent and forcing the cyclization by a terminal aryl substituent (limiting scope at the same time) is well precedented.

Thus, although this is certainly nice work, I do not think that the high standards for publication in Nat. Commun. are met.

**Pr. Gwilherm EVANO**

Laboratory of Organic Chemistry
Service de Chimie et PhysicoChimie Organiques
Université libre de Bruxelles
Avenue F. D. Roosevelt 50 CP160/06
1050 Brussels
BELGIUM

+32 (0)2 650 30 57

Gwilherm.Evano@ulb.be

<http://chimorg.ulb.ac.be/>

Brussels, November 1st, 2021

Dear reviewers,

Enclosed please find our revised manuscript entitled "A General Synthesis of Azetidines by Copper-Catalyzed, Photoinduced anti-Baldwin Radical Cyclization of Ynamides" that we are submitting for publication in *Nature Communications*. We are grateful for your careful analysis of our manuscript and have spent the last three months revising it in order to try to address all points raised: we hope you will be convinced by our revised manuscript.

According to the reviewer's comments and suggestions, the following changes have been made:

Reviewer 1:

"The manuscript titled "A General Synthesis of Azetidines by Copper-Catalyzed Photoinduced anti-Baldwin Radical Cyclization of Ynamides" by Jacob, Baguia, Dubart, Oger, Thilmany, Beaudelot, Deldaele, Perusko, Michelet, Neale, Romero, Moucheron, Van Speybroeck, Theunissen, and Evano describes the development of a radical cyclization approach to azetidines via a regioselective 4-exo-dig cyclization of nitrogen-substituted alkynes. The introduction to the manuscript clearly defines the new fundamental discovery in terms of cyclization technology, as well as the need for more selective ways to prepare and install azetidines. The scope and limitations of the new method are nicely described and the experimental procedures for starting material preparation and the highlighted cyclization are comprehensively covered in the Supporting Information. The computational studies provide kinetic reasoning to support the observed experimental trends and some of the derivatization studies, such as isomerization to the E-isomer, are well-selected."

- *"This manuscript describes a useful new synthetic process with interesting mechanistic implications that has been presented in a scholarly manner and merits publication in a top tier journal such as J. Am. Chem. Soc. or Angew. Chem. Int. Ed. However, the manuscript falls somewhat short of the benchmarks of the wide-readership of Nat. Comm. due to the lack of examples of the new method being used to access specific scaffolds found in target molecules or tested on more complex starting materials with potential competing reactivity. Addition of these examples, could make the manuscript suitable for publication in Nat. Comm.; otherwise, a more focused top tier chemistry journal is more appropriate."*

⇒ We do fully understand this remark and even if we had shown in our study of the scope and limitations that a wide array of azetidines with various substitution patterns and functional groups could be obtained, we have tried to bring this cyclization one step further and expand it to more complex substrates and to access specific scaffolds found in natural products and

drugs. Three of such complex/drug-like azetidines have been added to the manuscript and the corresponding paragraph has been added: "In order to push forward and further test the efficiency of this novel route to azetidines in "real-life" situation, the radical cyclization was performed from much more complex substrates (Fig. 4). In this perspective, phenylalanine- and estrone- containing ynamides **1ag** and **1ah** were subjected to our standard reaction conditions: even with these complex and challenging substrates, the radical cyclization proceeded efficiently, structurally complex azetidines **2ag** and **2ah** being obtained in 63 and 59% yields, respectively, showcasing the robustness of the cyclization. Gratifyingly, the cyclization is also amenable to the synthesis of spirocyclic azetidines such as **2ai** whose spirocyclic pyrrolidine-azetidine core can be found in diverse natural products and active pharmaceutical ingredients such as the chartellines **12**,³⁸ the chartellamides **13**³⁹ and delgocitinib **14**,⁴⁰ a janus kinase inhibitor developed by Japan Tobacco and recently commercialized for the treatment of autoimmune disorders and hypersensitivity, including inflammatory skin conditions.". The figure below has been added.

With these additional examples, we hope that we have addressed this point and managed to further highlight both the broad scope and potential of our method.

- "The cascade reaction shown in Figure 4 (now figure 5) is a highlight of the manuscript but could be more informative if the authors also disclosed the corresponding observed competing reactivity. It is important that the authors are able to isolate 23% of the single product but it would be interesting to know what other pathways are accessible and present opportunities for further optimization."

⇒ We have attempted this reaction five more times and have done our best to both optimize it and detect any byproduct formed. Reducing the reaction time improved the yield to 39% (the bicyclic product formed being not so stable under the reaction conditions) and in all trials, careful analysis of the crude reaction mixtures revealed that the bicyclic azetidine was the only product detectable, the low mass balance being attributed to competing oligomerization of the starting material. The text has been modified accordingly to clarify this point: "While the reaction was found to be moderately efficient and required a shorter reaction time, due to the limited stability of the bicyclic product formed, to reach 39% yield, a single cyclized product **2aj**, resulting from an unprecedented 4-exo-dig/7-endo-trig cascade radical cyclization - the latter having also little precedents due to competing 6-exo-trig cyclization⁴¹ - could be isolated (Fig. 5). Byproducts resulting from other cyclization modes or competing reactivity could not

be detected and the low mass balance was attributed to competing radical oligomerization of the starting ynamide."

- *"Is the iodine-promoted E/Z isomerization general for a range of azetidine products? The moderate dr is one of the drawbacks of Figure 3 but if the iodine isomerization is a general solution, this could address the alkene dr concern regarding Figure 3."*
 - ⇒ The moderate diastereoselectivity is indeed the main drawback of our cyclization and, as noted by this reviewer, this could be addressed if the iodine-promoted isomerization could be general. In this perspective, this iodine-promoted isomerization has been optimized (molecular sieves was added and the reaction was performed under more dilute conditions) to be as general as possible and a series of azetidines obtained as E/Z mixtures have been submitted to this isomerization: to our delight, this isomerization was found to be general. The following sentence has been added in the manuscript: "This isomerization was moreover shown to be general since azetidines **2d**, **2e**, **2g**, **2l**, **2aa**, **2ab**, **2ag** and **2ah** could all be fully isomerized to their E isomers in yields ranging from 80 to 88% (see Supporting Information for details)."

- *"The authors may want to reconsider the use of "peculiar" in the introduction. While it is unlikely that the authors intended the somewhat negative connotation of "peculiar", substitution with "specialized" or "specific" may be more appropriate in describing the work of Malacria and Kambe."*
 - ⇒ We did not indeed intend to mean anything negative about the work of Malacria and Kambe and we thank this reviewer for spotting this. "Peculiar" has been replaced by "specific".

- *"The spectra included in the Supporting Information is extensive, but many of the ¹H NMR spectra of the azetidines have obvious impurities in the baseline. For a large subset of these compounds, it seems like the concentration of the NMR sample is particularly dilute and that impurities in the NMR solvent rather than the compound itself are being observed. The authors should retake the ¹H NMR spectra for compounds such as **2a**, **2d**, **2ab**, where the baseline is indicating a dilute sample, and perform additional purification of compounds such as **2e**, **2k**, **2aa**, where the impurities seem more likely to be coming from the sample rather than the solvent."*
 - ⇒ As noted by this reviewer, we have provided the NMR spectra of an especially important number of new products in the Supporting Information. For most compounds, they have good to very good purities, but we agree that for some of them, some impurities could be found in the NMR spectra. As suggested by the reviewer, compounds **2e**, **2k** and **2aa** have been re-synthesized and/or repurified and we are now providing cleaner NMR spectra. Moreover, compounds (**Z**)-**2a**, (**E**)-**2a**, **2d**, **2g**, **2j**, **2k**, **2l**, **2ab** and **19** have also been re-synthesized and/or repurified to make sure all NMR spectra provided as Supporting Information would be as clean as possible and we hope the reviewers will appreciate this effort.
Copies of NMR spectra provided in the original Supporting Information and in the revised Supporting Information can be found in the annex to this letter (pages 9 to 30).

Reviewer 2:

"This manuscript by Jacob and coworker reports a new methodology of making azetidines allowing for a copper/photoredox catalysis via the challenging 4-exo-dig radical cyclization. The reaction proceeds with N-iodoethyl-ynamide and furnishes the corresponding azetidine products including further functionalization in fair to good yield. A wide range of scope is provided, and the methodology is extended to radical cascade cyclization via an interesting 4-exo-dig/7-endo-trig, and a single cyclized product is afforded. Furthermore, a deep mechanistic rationalization involving DFT computational work was conducted to support the regioselectivity of the radical cyclization. This is a significant progress in copper/photoredox catalysis radical cyclization from both synthetic and computational standpoints. Therefore, I recommend its publication in Nat. Comm. after revisions. Specific comments and questions are provided below:"

- "The reactions seem to work only with N-iodoethyl-ynamide and failed when replacing the protected amine by various methylene groups. What happens if replacing by dimethylmalonate?"

⇒ The calculations show that the cyclization is favored by the presence of the nitrogen atom in the starting ynamides and that the resulting vinyl radical species is also stabilized by the presence of the nitrogen atom. Replacing the protected amine by various methylene groups (CH₂, CHMe and CMe₂) indeed results in competing reduction and traces of cyclized products could not be detected in the crude reaction mixtures. We have chosen to study all CH₂, CHMe and CMe₂ groups to make sure the lack of cyclization would not come from a steric effect. The substrate mentioned by this reviewer would have been an interesting additional example but however, the starting material that would be required to study the influence of a malonate simply cannot be prepared. Indeed, a single procedure is reported for the synthesis of malonates substituted by both an alkyne and a precursor of the iodoethyl moiety (*Angew. Chem. Int. Ed.* **60**, 9706 (2021)) but, despite all our trials summarized in the scheme below, this procedure failed to provide the substrate that would have been required to study the influence of the malonate. Since this is not purely the focus of this manuscript and it would have required the development of an entirely new procedure for its synthesis, we have decided not to pursue with the preparation of this substrate that is not possible to synthesize with all methods available to date. Such alkynylated malonates are moreover known to easily cyclize to the corresponding furans (*J. Am. Chem. Soc.* **134**, 5766 (2012)), hence their limited stability.

- "The reactions work great with aromatic-substituted (R₃) ynamide. What happen with other radical stabilized groups, such as ester- and boronate-substituted, etc.?"

⇒ The cyclization indeed works best when starting from aryl-substituted ynamides due to the stabilization of the vinyl radical species resulting from the 4-exo-dig radical cyclization and this is one limitation of our process. An aromatic group is however not strictly required since a TIPS-substituted ynamide could also be cyclized, although with reduced efficiency (25% yield, as shown below, compound **2q** in Figure 3).

In addition to this substrate, many attempts have been made during the revision of this manuscript to prepare additional substrates. First, we know from our experience with ynamides that boron-substituted ones are unstable, so we discarded this option. As for ester-substituted ynamides, we have basically tried all procedures reported to date to prepare such ynamides but none of these procedures enabled the synthesis of such ynamides bearing a *N*-iodoethyl side chain (or precursors). Some of our attempts are shown in the scheme below (only optimized procedures are shown for clarity): all routes involve the synthesis of protected *N*-hydroxyethyl-ynamides, and such compounds spontaneously cyclize by an oxa-Michael addition upon deprotection.

The only substrate reported (*Tetrahedron Lett.* **26**, 4141 (1985)) with the appropriate substitution pattern is the one shown below: we have synthesized it but as expected, its cyclization is ineffective due to the highly strained nature of the bicyclic β -lactam-azetidine that would be formed. We cannot conclude that the cyclization is not effective when the ynamide is substituted by an ester group based on this really specific example only, so this example has not been included in our revised manuscript. Unfortunately, the synthesis of this ester-substituted *N*-iodoethyl-ynamide is restricted to this specific ynamide only and we could not extend it to other ester-substituted ynamides.

We have also tried to prepare a nitrogen-substituted ynamide (an “yndiamide”) by a procedure reported in 2017 (*Angew. Chem. Int. Ed.* **56**, 144428 (2017)) but such yndiamides with a *N*-iodoethyl side chain (or precursors) and mesyl groups cannot be prepared.

- "In Figure 3, the authors said the radical cyclization was inhibited with the presence of cyclopropyl group (**2s**). Did the authors observe any cyclization/ring-opening product? Although the authors showed product **2af** from compound **1af** and **1af'** afforded the same diastereoselective, the ring opening is also a strong evidence to support the radical pathway."

- ⇒ The trial with the substrate bearing a cyclopropyl group was indeed performed with the idea of promoting a radical cyclization/ring opening sequence that could favor the overall cyclization. Despite all attempts, only extensive degradation was observed in this case and products resulting from a radical cyclization/ring opening could not be detected. This is now mentioned in the manuscript: "Switching to non-aromatic substituents revealed the requirement for a radical-stabilizing group on the starting ynamide, TIPS-substituted azetidine **2q** being obtained in reduced yield (25%) while the presence of ethyl or cyclopropyl groups totally inhibited the cyclization, in the latter case, products resulting from a radical cyclization/ring opening being undetectable in the crude reaction mixture."
- Regarding the radical pathway, the diastereoselectivity of the cyclization performed from diastereoisomers **1af** and **1af'** yielding to the same product **2af** with close to identical stereoselectivities is, as pointed out by this reviewer, an evidence of a radical pathway that is moreover commonly accepted for such photoredox processes. Our work previously published on the photoredox activation of aryl/alkyl halides with the exact same photoredox catalyst (*Org. Lett.* **19**, 3576 (2017)) moreover strongly supports a radical pathway.
- "The authors should explain how to determine the stereochemistry of compound **(Z)-2a**, **(E)-2A**, and **2b**, etc. by NOESY in SI."
 - ⇒ NOESY experiments have indeed been used to determine the stereochemistry of compounds **(Z)-2a**, **(E)-2a** and **2b** as mentioned in the Supporting Information on pages S356 for **(Z)-2a**, S357 for **(E)-2a** and S358 for **2b**. Similarly, NOESY experiments have also been used to determine the *cis* configuration of the major isomer obtained after hydrogenation of azetidines **2t** into **15f**, **2v** into **15h** and **2aa** into **15i** as mentioned in the Supporting Information (page S359 for **15f**, S360 for **15h** and S361 for **15i**). A summary of all important correlations was also systematically presented at the end of the description of each compound in the Supporting Information.
 - "The authors should provide ¹⁹F NMR in SI"
 - ⇒ ¹⁹F NMR spectra are now provided for all compounds bearing fluorine atoms, i.e., **S1n**, **S2n**, **1n**, **2n** and **15e** (copies of ¹⁹F NMR spectra provided in the Supporting Information on pages S180 for **S1n**, S218 for **S2n**, S260 for **1n**, S306 for **2n** and S343 for **15e**).
 - "The NMR of compounds **1af'** and **16** look impure"
 - ⇒ As mentioned above, we have re-synthesized and/or repurified many compounds to make sure all NMR spectra provided as Supporting Information would be as clean as possible: these include compound **16**. As for compound **1af'**, this compound is quite unstable, and we could not obtain a cleaner NMR spectrum for this intermediate.
 - "Based on the computed energetics for 4-exo-dig cyclization, **1r** has a similar barrier as **1a**, and is also thermodynamically favorable, albeit with smaller thermodynamic drive (-5.3 vs -9.8 kcal/mol). Meanwhile, the 5-endo-trig cyclization of **1r** is even smaller in barrier (16.5 kcal/mol) compared to **1a** (17.1 kcal/mol), and the resulting products are in similar energy levels. These results seem unable to explain the experimental results of **1r** being inactive or undergoing reduction instead of cyclizations. What is the energetics for **4P_{1r}** undergoing following reduction? This might provide some insight into the reactivity of **1r**."
 - ⇒ We thank the reviewer for this constructive comment. We have computed the barrier for reduction of **4P_{1r}**, labelled **TS-P_{1r}** (+6.8 kcal/mol) in the figure below. The energetic span is +12.1 kcal/mol and so is lower than for reduction of **4P_{1a}**, suggesting that if formation of **4P_{1r}** is

accessible, onwards reduction should be too. This is not aligned with experiment as no 4-exo-dig cyclization is observed for **1r**, therefore the hindering of observed cyclization appears to lie elsewhere in the pathway. To further explore this we have computed the reductions of **I**_{1a}, **I**_{1r} and **I**₈ (the barriers for 4-exo-dig cyclization are in parentheses for comparison):

1a: $\Delta G^\ddagger = +15.8$ (+13.4): **TS-R**_{1a}

1r: $\Delta G^\ddagger = +15.4$ (+14.7): **TS-R**_{1r}

8: $\Delta G^\ddagger = +19.0$ (+16.3): **TS-R**₈

4-exo-dig cyclization is still kinetically favored in **1r** ($\Delta\Delta G^\ddagger = 0.7$ kcal/mol), and so these energetics suggest that 4-exo-dig should indeed be observed for **1r**. We considered whether a more appropriate chemical model could better describe the energetics of reduction. When ⁱPr₂NEt is employed, the corresponding cationic radical ⁱPr₂NEt^{•+} is often implicated as the reductant, and so the barriers for these reduction processes were re-computed using ⁱPr₂NEt^{•+} as the reductant whereby the TSs are in an overall S=1 state, and this included fully repeating the conformer searching procedure in each case as a change in the overall spin-state of the system led to alternate TS geometries. The activation barriers were found to increase in this model (the figure below contains the new transition state energies, underlined, beside the previously computed barriers). While now higher in energy than for **TS-P**_{1r}, the new S=1 barrier is still low enough to suggest kinetically accessible 4-exo-dig cyclization for **1r**, and moreover the reduction of **I**_{1r} is still disfavored with respect to cyclization and onwards reduction, where, for example, ³TS-R_{1r} = +22.2 kcal/mol. We would like to stress, as we had done in the manuscript, that the original chemical model employed for the reduction in the calculations was chosen simply to assess the steric competition between E:Z alkene formation in **1a**. We are confident that it can reliably be used in this vein to assay the sterics of intermolecular HAT at each face of **4P**_{1a} but anticipate that it is too approximate, with respect to experiment for optimal description of the quantum mechanical effects of hydrogen atom transfer (such as PCET). For this purpose, more advanced levels of theory that are beyond the scope of this investigation (and would require a study in its own right) should be considered, such as CASPT2 with Marcus theory (*ACS. Catal.* **8**, 7388 (2018) or ring polymer molecular dynamics (*J. Chem. Phys.* **138**, 134109 (2013)).

Note: all new stationary points computed can be found in the Supporting Information.

Reviewer 3:

“The authors present a well-conducted study on the photochemical cyclization of ynamides that lead to azetidines. The results obtained are certainly an interesting contribution to the growing toolbox of light-induced, radical cyclizations, and moreover, a useful class of compounds is accessible by this approach. The supporting information makes a good impression, giving credit to the work.

Nevertheless, I am not convinced that the study presented meets the requirements for publication in Nat. Commun. By now it is recognized that 4-exo-dig cyclizations cannot really be considered to be an exception to the Baldwin rules (work by Gilmore). Moreover, the work by Kambe (reference 18) reports a quite related cyclization leading to very similar products as reported in this study. The design principle by the authors, i.e. biasing the conformation by an N-substituent and forcing the cyclization by a terminal aryl substituent (limiting scope at the same time) is well precedented.

Thus, although this is certainly nice work, I do not think that the high standards for publication in Nat. Commun. are met.”

- ⇒ We thank this reviewer for assessing our manuscript and we have heard his opinion. However, and as mentioned in our manuscript, 4-exo-dig cyclizations are rare and radical ones even rarer (one example by the Malacria group and one by the Kambe group for which we must argue that the nature of the radical species (carbamoyl radicals vs alkyl radicals in our case) is totally different, as is the nature of the final products (α -telluromethylene- β -lactams vs 2-benzylideneazetidines in our case)). Even with the work by Gilmore and Alabugin, which suggests that radical 4-exo dig cyclizations are feasible, there is no general and experimental study of such cyclizations and competing 5-endo-dig processes are also reported in the literature.

Based on all the literature reported to date, a detailed experimental study on such 4-exo-dig cyclizations is still missing and our manuscript aims to contribute filling this gap, which should bring it to the high standards for publications in *Nature Communications*.

Moreover, our work also provides important contributions to the synthesis of azetidines, building blocks of growing interest that are not that easy to prepare, and to copper-based photoredox catalysis, which is still much less developed compared to iridium and ruthenium-based photoredox catalysis despite an enormous potential.

Hoping to have addressed all points raised by the reviewers and that this revised manuscript will be suitable for publication in *Nature Communications*.

Yours sincerely,

Gwilherm EVANO

Annex:

*copies of NMR spectra in the Supporting Information originally provided (“before”)
and provided in the revised Supporting Information (“after”)*

(Z)-2a

Before

After

Before

After

Before

After

2d

Before

After

Before

After

2e

Before

After

Before

2e

After

Before

After

Before

After

Before

After

Before

After

Before

After

Before

2k

After

Before

After

Before

After

Before

After

Before

After

Before

After

Before

After

19

Before

After

19

Before

After

Reviewers' Comments:

Reviewer #1:

Remarks to the Author:

The manuscript titled "A General Synthesis of Azetidines by Copper-Catalyzed Photoinduced anti-Baldwin Radical Cyclization of Ynamides" by Jacob, Baguia, Dubart, Oger, Thilmany, Beaudelot, Deldaele, Perusko, Landrain, Michelet, Neale, Romero, Moucheron, Van Speybroeck, Theunissen, and Evano has been revised to address previous comments by this reviewer. Notably, examples of more complex substrates have been shown to tolerate the cyclization conditions including ynamides that give spirocyclic azetidine motifs found in natural products. The iodine catalyzed E/Z-isomerization has also been tested with additional substrates, which indicate the generality of this solution for consolidating the alkene mixtures generated from the cyclization reaction into a primarily single isomer. The purity of the compounds in the Supporting Information has been improved to appropriate levels for publication and small wording changes have been fixed. The authors also put in a significant amount of effort to address and understand the cascade reaction in Figure 5; however, the authors' dual reasoning that the low yield is due to starting material oligomerization, while the azetidine product is also described as unstable under reaction conditions, seems inconsistent. Can the authors distinguish the difference between product decomposition and azetidine decomposition? Overall, the authors have clearly addressed this reviewer's initial criticisms of the manuscript and the article has been revised appropriately for publication in Nat. Commun.

Reviewer #2:

Remarks to the Author:

I think that the revised manuscript and SI are good and suitable for publication in Nature Communications, especially considering these further reaction scope investigations, further DFT studies and corresponding discussions why the experimental results of 1r being inactive or undergoing reduction instead of 4-exo-dig cyclization. However, the calculation result of 1r still can't explain the experimental result. It shows that alkyl-substituted ynamide could be cyclized. I suggest that the authors should move the calculation of 1r (green pathway) to the SI including the corresponding discussions and keep the assessments of the steric competition between E:Z alkene formation in 1a in the manuscript. In my opinion, the transformation of alkyl-substituted ynamide is still a good study if involves a suitable reductant.

**Pr. Gwilherm EVANO**

Laboratory of Organic Chemistry
Service de Chimie et PhysicoChimie Organiques
Université libre de Bruxelles
Avenue F. D. Roosevelt 50 CP160/06
1050 Brussels
BELGIUM

+32 (0)2 650 30 57

Gwilherm.Evano@ulb.be

<http://chimorg.ulb.ac.be/>

Brussels, December 14th, 2021

Dear reviewers,

Enclosed please find our revised manuscript entitled “A General Synthesis of Azetidines by Copper-Catalyzed, Photoinduced anti-Baldwin Radical Cyclization of Ynamides” that we are submitting for publication in *Nature Communications*. We are grateful for your second analysis of our manuscript and have revised our manuscript according to your comments: we hope that our manuscript will now be suitable for publication in *Nature Communications*.

According to the reviewer’s comments and suggestions, the following changes have been made:

Reviewer 1:

- *The manuscript titled “A General Synthesis of Azetidines by Copper-Catalyzed Photoinduced anti-Baldwin Radical Cyclization of Ynamides” by Jacob, Baguia, Dubart, Oger, Thilmany, Beaudelot, Deldaele, Perusko, Landrain, Michelet, Neale, Romero, Moucheron, Van Speybroeck, Theunissen, and Evano has been revised to address previous comments by this reviewer. Notably, examples of more complex substrates have been shown to tolerate the cyclization conditions including ynamides that give spirocyclic azetidine motifs found in natural products. The iodine catalyzed E/Z-isomerization has also been tested with additional substrates, which indicate the generality of this solution for consolidating the alkene mixtures generated from the cyclization reaction into a primarily single isomer. The purity of the compounds in the Supporting Information has been improved to appropriate levels for publication and small wording changes have been fixed.*
- *The authors also put in a significant amount of effort to address and understand the cascade reaction in Figure 5; however, the authors’ dual reasoning that the low yield is due to starting material oligomerization, while the azetidine product is also described as unstable under reaction conditions, seems inconsistent. Can the authors distinguish the difference between product decomposition and azetidine decomposition? Overall, the authors have clearly addressed this reviewer’s initial criticisms of the manuscript and the article has been revised appropriately for publication in Nat. Commun.*
 - ⇒ The rationale for the low yield obtained was indeed not crystal clear and a bit confusing. We have therefore attempted this reaction one more time and we increased the reaction time to probe the bicyclic azetidine’s stability. Increasing the reaction time from 16 to 60 hours afforded a similar yield in azetidine (38% obtained after 60 hours compared to 39% obtained after 16 hours) which is indicative that the bicyclic azetidine is actually quite stable in the

reaction conditions. Once again, a careful analysis of the crude reaction mixture revealed that the bicyclic azetidine was the only product detectable. The text has been modified accordingly: “While the reaction was found to be moderately efficient and required a shorter reaction time to reach 39% yield, which did not improve after 16 hours, a single cyclized product **2aj**, resulting from an unprecedented 4-exo-dig/7-endo-trig cascade radical cyclization - the latter having also little precedents due to competing 6-exo-trig cyclization⁴¹ - could be isolated (Fig. 5). Byproducts resulting from other cyclization modes or competing reactivity could not be detected and the low mass balance was attributed to competing radical oligomerization of the starting ynamide.”

Reviewer 2:

“I think that the revised manuscript and SI are good and suitable for publication in Nature Communications, especially considering these further reaction scope investigations, further DFT studies and corresponding discussions why the experimental results of **1r** being inactive or undergoing reduction instead of 4-exo-dig cyclization.

- However, the calculation result of **1r** still can't explain the experimental result. It shows that alkyl-substituted ynamide could be cyclized. I suggest that the authors should move the calculation of **1r** (green pathway) to the SI including the corresponding discussions and keep the assessments of the steric competition between *E*:*Z* alkene formation in **1a** in the manuscript. In my opinion, the transformation of alkyl-substituted ynamide is still a good study if involves a suitable reductant.”

⇒ We do fully agree with the reviewer and we moved the calculation of **1r** (red pathway in fact, not the green one) and the corresponding discussion to the Supporting Information. Figure 7 and the corresponding discussion in the manuscript have been modified accordingly: “Characterization of 4-exo-dig vs 5-endo-dig cyclizations for compound **8** revealed a qualitatively consistent kinetic and thermodynamic scenario with that of ynamide **1a**, where the 4-exo-dig cyclization is kinetically favoured over the 5-endo-dig cyclization, the latter remaining however the thermodynamically favoured process. Importantly, the barrier for the 4-exo-dig cyclization of ynamide **1a** is lower than that of compound **8**, which nicely reflects the difference of reactivity observed experimentally between aromatic-substituted ynamides **1a-p,t-ad,af** and compounds **6-8**. Moreover, a comparison of the difference between 4-exo-dig and 5-endo-dig barriers in **1a** (where $\Delta\Delta G^\ddagger = 3.7$ kcal/mol) with that of **8** ($\Delta\Delta G^\ddagger = 3.9$ kcal/mol) reveals that the *N*-mesyl moiety equally accelerates both stereodivergent intramolecular cyclization processes, as opposed to more favourably enhancing the 4-exo-dig process.

To further probe the underlying cause of this difference in reactivity, electronic structure analyses of **1a** and **8** were undertaken to evaluate the influence of the triple bond polarization

on the outcome of the cyclization. While computed partial atomic charges somewhat reveal the polarized nature of the triple bond in ynamides **1a-ai** (see Supplementary Table 1), we also calculated the bond polarities according to the index introduced by Raub and Jansen (also referred to as the Raub-Jansen index). According to their definition, the index, which will hereafter be labelled as the bond-polarity index p_{xy} , calculates atomic contributions to electron densities of chemical bonds via complementary analyses of AIM and ELF electron density partitions. The triple bond polarity of **1a** ($p_{CC} = 0.17$) is stronger than that of compound **8** ($p_{CC} = 0.01$), as **1a** has the largest polarity index. This nicely reflects the reactivity of the substrates towards intramolecular radical cyclization and confirms that the nitrogen atom has the most significant polarizing effect, while the phenyl group only marginally contributes to bond polarization. The role of the aromatic group is most likely to stabilize the resulting transient vinyl radical prior to onwards reduction. Indeed, evaluation of residual spin-density in the resulting transient vinyl radical species **4P_{1a}** and **4P₈** (see Supplementary Information) clearly shows this role of the aromatic group, as residual unpaired spin density is drawn from the carbon atom α to the nitrogen atom and shared across alternating C atoms in the aromatic ring. Despite this role of the phenyl group in stabilizing **4P₈**, the lack of reactivity of **8** itself towards 4-exo-dig cyclization demonstrates however the synergistic requirement of both the nitrogen atom and the aromatic group to promote the 4-exo-dig radical cyclization.

In addition to these changes, all points listed in the Author Checklist have been addressed.

Hoping to have addressed all remaining points raised by the reviewers and that this revised manuscript will be suitable for publication in *Nature Communications*.

Yours sincerely,

Gwilherm EVANO